# Modification of the Linker Amino Acid in the Cell-Penetrating Peptide NickFect55 Leads to Enhanced pDNA Transfection for In Vivo Applications

**DOI:** 10.3390/pharmaceutics15030883

**Published:** 2023-03-09

**Authors:** Heleri H. Härk, Ly Porosk, Lucas R. de Mello, Piret Arukuusk, Emerson R. da Silva, Kaido Kurrikoff

**Affiliations:** 1Institute of Technology, University of Tartu, Nooruse 1, 50411 Tartu, Estonia; ly.porosk@ut.ee (L.P.); piret.arukuusk@ut.ee (P.A.); kaido.kurrikoff@ut.ee (K.K.); 2Departamento de Biofisica, Universidade Federal de São Paulo, São Paulo 04023-062, Brazil; lucasr.mello@hotmail.com (L.R.d.M.); er.silva@unifesp.br (E.R.d.S.)

**Keywords:** cell-penetrating peptides, in vivo transfection, pDNA delivery

## Abstract

Despite numerous efforts over the last three decades, nucleic acid-based therapeutics still lack delivery platforms in the clinical stage. Cell-penetrating peptides (CPPs) may offer solutions as potential delivery vectors. We have previously shown that designing a “kinked” structure in the peptide backbone resulted in a CPP with efficient in vitro transfection properties. Further optimization of the charge distribution in the C-terminal part of the peptide led to potent in vivo activity with the resultant CPP NickFect55 (NF55). Currently, the impact of the linker amino acid was further investigated in the CPP NF55, with the aim to discover potential transfection reagents for in vivo application. Taking into account the expression of the delivered reporter in the lung tissue of mice, and the cell transfection in the human lung adenocarcinoma cell line, the new peptides NF55-Dap and NF55-Dab* have a high potential for delivering nucleic acid-based therapeutics to treat lung associated diseases, such as adenocarcinoma.

## 1. Introduction

Nucleic acid-based therapeutics may offer treatment opportunities for various diseases at the transcriptional and translational levels [1]. However, intracellular delivery of nucleic acids is limited due to their large molecular weight and polarity. Additional challenges such as low in vivo stability and rapid enzymatic degradation limit the use of nucleic acids for therapeutic purposes [2]. In order to overcome these issues, an efficient delivery system is required to deliver the therapeutic nucleic acids into the target cells, tissues, and organs.

The research of the past 30 years has led to the discovery of safe and efficient delivery vectors for nucleic acids including viral vectors, lipid nanoparticles, cationic polymers, and inorganic nanoparticles [1,2]. Despite the numerous delivery vectors available for different applications, only a few of them have managed to reach the clinical stage. For example, siRNA formulations in LNP or with GalNAc are in clinical use [3]. However, these methods are specific to hepatocytes, whereas delivering into extrahepatic tissues remains an unresolved challenge in the field [4]. Hence, there is an urgent need for efficient delivery systems especially for in vivo applications. One of the potential delivery vector types for in vivo applications is the cell-penetrating peptides (CPPs), which are relatively short peptides, consisting of approximately 5–30 amino acid residues. CPPs have been used as nonviral delivery vectors for the intracellular delivery of various bioactive cargos, including nucleic acids [5,6,7,8,9]. Several CPP-based molecules with therapeutic potential have reached different stages of clinical trials [10], and the CPP-based neuromodulator Daxxify has been approved by the FDA. A major advantage of CPPs is their ability to enter the cells in a noninvasive way. CPPs are therefore considered a safe and efficient delivery platform for therapeutic molecules [11].

In our previous efforts, we have focused on designing nucleic acid delivery methods for in vivo applications and we have introduced the NickFect (NF) series of CPPs which are proven to be efficient for gene delivery in vivo. During the sequential development of CPP iterations, we have shown that altering the amino acid in the seventh position of Stearyl-TP10 (PepFect3) linking together neuropeptide galanin motif and mastoparan residues led to an improved nucleic acid delivery vector [6]. One of the key findings was that replacing the “linker” amino acid Lys_7_ with Orn_7_, and continuing synthesis via the ε-NH_2_ group instead of α-NH_2_ of the linker amino acid Ornithine resulted in a CPP with a significant rise of the pDNA transfection efficiency for in vitro applications [7,8]. In further works, we increased the in vivo delivery efficacy of the CPP by optimizing the charge distribution in the C-terminal part of the peptide sequence. We introduced the CPP NF55 [8], which achieved potent in vivo delivery of pDNA and siRNA [9]. This enabled us to create a platform for the delivery of nucleic acids which could contribute to the clinical development of nucleic acid-based therapeutics.

The aim of our current work was to further explore the importance of the linker region in the CPP NF55 in the delivery of a pDNA type nucleic acid in vivo. We compared alternative linker regions in the parent NF55 sequence with the aim of obtaining optimal pDNA condensation into nanoparticles and enhanced nucleic acid delivery in vivo. For this, we (a) used amino acids with different side chain lengths as linker amino acids and (b) compared the effect of the synthesis from the α-amino group (“linear” peptides) versus the side chain amino group (“kinked” peptides) of the linker amino acid.

## 2. Materials and Methods

### 2.1. Peptide Synthesis

Peptides were synthesized on an automated peptide synthesizer Biotage Initiator^+^ Alstra (Biotage, Uppsala, Sweden) using the fluorenylmethyloxycarbonyl (Fmoc) solid phase peptide synthesis strategy. For the synthesis of peptides, where the chain is continued via the ε-NH_2_ group of the linker amino acid, Boc-L-Dap(Fmoc)-OH, Boc-L-Dab(Fmoc)-OH, Boc-L-Orn(Fmoc)-OH, and Boc-L-Lys(Fmoc)-OH (Iris Biotech GmbH, Marktredwitz, Germany) were used. The reaction was carried out in DMF by coupling the amino acid (5 eq) using HOBT and HBTU (5 eq) as coupling reagents and DIEA (10 eq) as an activator base to Rink amide ChemMatrix resin (Biotage, Uppsala, Sweden) with a loading of 0.41 mmol/g (0.125 mmol, 1 eq) which was used as a solid phase to obtain C-terminally amidated peptides. Stearic acid (5 eq) was coupled to the N-terminus of peptidyl resin manually in DCM using the same strategy as when coupling amino acids.

Cleavage was performed following the standard protocol with trifluoroacetic acid (TFA), 2.5% triisopropylsilane, and 2.5% ultrapure water for 2 h at room temperature. Peptides were purified by reversed phase high-performance liquid chromatography on a C3 column (Agilent Zorbax 300SB-C3, 5 μm, 250 × 9.4 mm) using a gradient of acetonitrile–water containing 0.1% TFA. The purity of peptides was confirmed by reverse-phase ultraperformance liquid chromatography using a C18 column (ACQUITY UPLC BEH130 C18, 1.7 μm, 100 × 2.1 mm) and a solvent system of acetonitrile (B)–water (A) containing 0.1% TFA with a gradient of B = 5–80%. The molecular weight of the peptides was analyzed by matrix-assisted laser desorption–ionization and time-of-flight mass spectrometry (Bruker Daltonics GmbH & Co. KG, Bremen, Germany). The specific concentration of the peptides was achieved by diluting accurately weighed substances and the absorption of tyrosine.

### 2.2. The Formation of CPP–pDNA Complexes

The CPP–pDNA complexes were prepared by mixing plasmid DNA (0.5 μg of EGFP encoding pEGFP (size 4731 bp; Clontech Laboratories Inc., Montain View, CA, USA ) per well for 24-well plate transfection or 0.1 μg firefly luciferase encoding pLuc (size 10,060 bp) per well for a 96-well plate) with CPPs at a CPP–pDNA charge ratio (CR) of 1:1 to 4:1 (if not indicated otherwise) in MilliQ water, followed by incubation for 30 min at room temperature. CR was calculated theoretically considering the net positive charges of the peptide and the negative charges in the pDNA backbone.

PEI MAX (MW = 40,000; Polysciences, Inc., Warrington, PA, USA) was mixed with plasmid DNA in MilliQ water at the indicated N/P ratio. Thereafter, a complex solution was incubated for 20 min at room temperature.

For in vivo studies, the pLuc was mixed with the CPP at CR2 in MilliQ water, wherein the dose per animal was 2.5 mg/kg. After 30 min of incubation at room temperature, glucose was added to the complexes to obtain an isotonic solution with a final injection volume of 336 µL, which was immediately injected intravenously via the tail vein.

### 2.3. Characterisation of the CPPs

The secondary structure of the peptides was determined by CD spectroscopy using a Chirascan CD spectrometer (Applied Photophysics Ltd., Leatherhead, UK). For measurement, peptides in ultrapure water with a concentration of 100 μM were transferred to a quartz cuvette (Hellma GmbH & Co. KG, Müllheim, Germany) with an optical path length of 1 mm. The signal was recorded for wavelength intervals between 185 nm and 260 nm, using a 1 nm bandwidth. Based on the obtained CD spectra, the secondary structures of the peptides in water were calculated using BeStSel secondary structure prediction server [12].

### 2.4. The Assessment of ξ-Potential of the CPP–pDNA Complexes

For the measurements of ξ-potential, dynamic light scattering studies were performed using a Zetasizer Nano ZS apparatus (Malvern Panalytical, Malvern, UK). Complexes between the pLuc and the CPP were formed as described previously and diluted 10-fold prior to the measurements.

### 2.5. The Assessment of the CPP–pDNA Complex Formation by Quantitation of Nucleic Acid Intercalating Dye, the Stability of the Complexes towards Enzymatic Degradation by Proteinase K, and pDNA Displacement from the Complexes by Heparin Sodium Salt

To evaluate the complexation of pDNA by the CPP and the stability of formed complexes to enzymatic degradation by Proteinase K, we quantified the accessible pDNA with Quant-iT PicoGreen (PG) (Thermo Fisher Scientific, Vantaa, Finland). Briefly, the preformed CPP–pLuc complexes were transferred to a black opaque 96-well plate and diluted with 115 µL of MilliQ water. Thereafter, 10 µL of diluted fluorescent DNA intercalating dye (Quant-iT PicoGreen) was added to the complexes and after 5 min of incubation, the initial accessibility of pDNA was quantified by measuring the fluorescence (λ_ex_ = 492, λ_em_ = 535). For assessing the stability of the complexes towards enzymatic degradation, 10 µL of Proteinase K (>21 U) was added after ensuring the formation of complexes. The fluorescence was measured over an indicated period at 25 °C and the values were normalized against the fluorescence of free (uncomplexed) pDNA at the same concentration as used for the complexes. For assessing the stability of the complexes against the displacement of pDNA by heparin, the preformed CPP–pDNA nanoparticles were coincubated with different concentrations of heparin sodium salt, and measured fluorescence was normalized against the fluorescence of free pDNA.

### 2.6. The Assessment of the CPP–pDNA Complex Formation by Gel Migration Assay

To investigate the ability of the NF55 analogues to interact with the pDNA and limit its migration in agarose gel, complexes were formed between the NF55 analogues and pLuc (size 4731 bp), as indicated in Section 2.2 at CR2. After incubation, samples were diluted with MilliQ water 2 times, and the loading dye was added. The samples were transferred to an agarose gel tooth. Gel electrophoresis was performed in agarose gel (1% agarose gel in 1× TAE buffer) in 1× TAE buffer at 80 mA for 1 h. The accessible pDNA was visualised under UV light using ethidium bromide. As a ladder, ZipRuler Express DNA ladder 2 was used (Thermo Fisher Scientific, Vantaa, Finland).

### 2.7. The Analysis of the Nanoscopic Structure of the CPP–pDNA Complexes by Atomic Force Microscopy Combined with Infrared Spectroscopy (AFM-IR)

Spatially-resolved infrared measurements were carried out using a combination of atomic force microscopy and nanoscale infrared spectroscopy on an Anasys NanoIR2-s instrument. Complexes between CPP and pLuc were formed at CR2 as described in Section 2.2. Droplets from the complex mixtures were then placed onto Au-coated substrates, which were dried under a mild vacuum in a desiccator. Data collection was carried out in contact mode by scanning surface areas of 2 × 2 μm^2^ or 3 × 3 μm^2^ at a resolution of 256 × 256 pixels. The cantilever was equipped with ContGB-G probes, which had a spring constant of 0.2 N/m and a tip radius of 25 nm. Infrared spectra were obtained by positioning the probe on the top of a specific nanoparticle and illuminating the substrate with laser light in the infrared range of 1570–1800 cm^−1^. Initial image processing was performed using Analysis Studio software version 3.14, which eliminated noise from the spectra using Fast Fourier transform filters (FFT, 10 points). Further visualization enhancements were carried out using Gwyddion software version 2.51 [13].

### 2.8. Cell Culture Maintenance

CHO-K1 and A549 cells were cultured in a humidified environment at 37 °C, 5% CO_2_, and cultivated in Dulbecco’s Modified Eagle’s Medium (DMEM) for the CHO-K1 cell line and in RPMI-1640 for the A549 cell line. Both media were supplemented with GlutaMax, 0.1 mM nonessential amino acids, 1.0 mM sodium pyruvate, 10% fetal bovine serum (FBS), 100 U/mL penicillin, and 100 μg/mL streptomycin.

### 2.9. The Transfection of pDNA in Cell Culture

For the plasmid delivery assay, 10,000 CHO-K1 or A549 cells per well were seeded in a serum-containing media 24 h prior to each experiment into a 96-well plate. The luciferase encoding plasmid pLuc was mixed with CPPs at the indicated CR, as described in Section 2.2 and the cells were incubated with complexes for 24 h in a media with serum or without serum. After incubation, the cells were washed with 1× PBS and lysed by treating the cells with 30 µL of 0.1% Triton X-100 in PBS at +4 °C for 30 min. Thereafter, 20 µL of cell lysate was transferred to a black frame and a white well 96-well plate and luciferase activity was measured using a GloMax 96 Microplate Luminometer (Promega Biotech AB, Stockholm, Sweden) after treatment with a luciferin-containing solution. Acquired results were normalized against total protein content, which was obtained by treating the remaining cell lysate with a Pierce BCA Protein Assay kit (Thermo Scientific, Vantaa, Finland) and determined by measuring the absorbance at 562 nm.

For flow cytometry analysis used to determine the transfected cell population, 50,000 CHO-K1 cells per well in serum-containing media were seeded 24 h prior to each experiment into a 24-well plate. The eGFP encoding plasmid was mixed with the CPPs at CR3 as described in Section 2.2 and cells were incubated with the complexes for 24 h in a media with serum or without serum. Following incubation, cells were washed with 1× PBS, detached from the plate using trypsin-EDTA (0.25%), resuspended in 1× PBS containing 1% FBS, and transferred to 1.5 mL tubes. Flow cytometry was performed using Attune NxT Flow Cytometer equipped with a 488 nm argon laser, where the population of viable cells was determined from a plot of forward scattered vs. side scattered light. Attune NxT Software 3.2.1 software was used to analyze a minimum of 10,000 events per sample from the viable cell population. Results were shown as a percentage of GFP-positive cells from the viable cell population.

### 2.10. The Quantification of Reporter Levels In Vivo

All animal experiments and procedures were approved by the Estonian Laboratory Animal Ethics Committee (approvals no. 110 12 June 2017 and no. 203 22 September 2021). For in vivo experiments, male and female, 8-week-old BALB/c mice were used.

Complexes were formed as described in Section 2.2. The expressed reporter gene levels were assessed by using bioluminescence live animal imaging IVIS (details under Section 2.11) 16 h after injecting the complexes. After imaging the mice were sacrificed, tissues were harvested, and bioluminescence was detected from the tissue homogenates (details under Section 2.12).

### 2.11. Bioluminescence Live Animal Imaging

To assess the expression levels of luciferase-encoding pDNA, in vivo bioluminescence imaging was performed using the IVIS Lumina II (PerkinElmer, Inc., Waltham, MA, USA). For this, a solution of D-luciferin (PerkinElmer, Inc., Waltham, MA, USA) in DPBS free of Mg^2+^ and Ca^2+^, which was sterile-filtered with 0.2 µm syringe filters prior to in vivo use, was administered by intraperitoneal injection, at the dose of 150 mg/kg. Throughout the imaging process, the mice were anaesthetized by using isoflurane (3% for induction, 1% for maintenance). Photon emissions from the live animals were quantified 10 min later with an exposure time of 1 min. Regions of interest (ROI) were quantified as average radiance (photons s^−1^ cm^−2^ sr^−1^), represented by colour scale (IVIS Living Image 4.0).

### 2.12. The Quantification of Luciferase Activity from the Tissues Ex Vivo

To evaluate the amount of luminescence from the tissue homogenates, the whole tissues were first lysed by adding 700 µL of T-PER Tissue Protein Extraction Reagent (Thermo Scientific, Vantaa, Finland) followed by homogenization using Precellys24 Tissue Homogenizer (Bertin Corp., Rockville, MD, USA). Thereafter, the samples were shaken for 15 min at 3000 rpm and then centrifuged (3 min at 10,000× *g* and at +4 °C). The supernatant was transferred to a clean tube and 500 µL of the lysis buffer was added to the pellet, followed by shaking and centrifugation. The supernatant was collected and mixed with the previous one, followed by gentle mixing. From this, 20 µL were transferred to a black frame white well 96-well plate and the amount of luciferase was measured by using the Promega luciferase assay system according to the manufacturer’s protocol in combination with GloMax 96 Microplate Luminometer (Promega Biotech AB, Stockholm, Sweden). An average RLU (relative light unit) of two technical replicates was normalized against the total protein content, which was obtained by treating 10 µL of 20× diluted tissue lysate with a Pierce BCA Protein Assay Kit (Thermo Scientific, Vantaa, Finland) and determined by measuring the absorbance at 562 nm.

### 2.13. The Assessment of Safety of the NF55 Analogues

Cell viability was assessed by using a CellTiter 96 AQ_ueous_ One Solution Cell Proliferation Assay (MTS) (Promega Biotech AB, Stockholm, Sweden) according to the manufacturer’s instructions. For this, 10,000 CHO-K1 cells per well were seeded 24 h prior to the experiment in a 96-well plate in a serum-containing media. Before transfecting the cells with the complexes (CPP and pLuc complexes formed at CR3), the media on the cells was changed for a phenol red-free, serum-free DMEM media. Per well, 20 µL of the MTS reagent was added 20 h post-transfection, and the absorbance of the formazan product was measured at 490 nm after 2–4 h incubation with the reagent. The results are shown as the percentage of viable cells normalized to untreated cells (the latter are defined as 100%).

In vivo safety was evaluated by measuring the alanine aminotransferase and the aspartate aminotransferase levels from the blood of mice injected with nanoparticles formed between NF55 analogues and pDNA. Additionally, the mice were weighed preinjection with the CPP-pDNA complexes and prior to sacrifice.

### 2.14. Statistical Analysis

The statistical analyses were done using GraphPad Prism version 9.3.1 (Graphpad Software, San Diego, CA, USA). The values in all experiments are shown as the mean ± SEM of 2–3 independent experiments including at least 3 replicates per experiment, if not indicated otherwise.

## 3. Results and Discussion

### 3.1. Design and Characterization of Peptides

We have previously shown that designing a kinked structure in the peptide backbone resulted in the CPP NF51, which demonstrated efficient in vitro transfection properties [6,7]. Further optimization of the C-terminal charge distribution of the peptide allowed potent in vivo activity of the resultant NF55 [8]. Currently, we aimed to explore if the in vivo activity of NF55 can be further improved by optimizing the linker region.

To assess the impact of modifying the linker amino acid on the delivery efficacy of the CPP, three other amino acids—Lysine (Lys), diaminobutyric acid (Dab), and diaminopropionic acid (Dap)—were used in the place of the parent amino acid ornithine (Orn). These amino acids differ by the number of methyl groups in their side chain. For each of the new linker peptides, two approaches were applied in parallel—prolongation of the sequence by coupling amino acid on the sixth position to the ⍺-NH_2_ group, creating a set of linear peptides, or coupling to the NH_2_ group in the side chain, creating a set of kinked peptides. Therefore, seven new peptides were designed and tested in addition to the previously known CPP NF55. The linear peptides differ from each other by linker amino acid side group length—in increasing order, Dap → Dab → Orn → Lys, each peptide has one more methyl group, which increases the side chain bulkiness (Figure 1a, Appendix A; the analogues without the asterisk in the peptide name). The kinked peptides have the same side chain but differ from each other by the length of the linker (Figure 1b, Appendix A; the analogues with the asterisk in the peptide name).

CPP interactions with the cell membranes have been associated with the helical secondary structure of the peptide [14] and a higher percentage of α-helicity in the sequence has a positive impact on in vivo delivery of nucleic acids [8]. Circular dichroism (CD) spectroscopy has been widely used to determine the secondary structure of peptides [15]. Therefore, as a first step, CD spectra of the peptides were detected to determine whether a modification in the linker position of NF55 results in conformational differences between the peptides. Based on the results obtained by CD spectroscopy, the secondary structures of the peptides in water were calculated (Figure 2b). All of the free peptides adopted a mostly α-helical structure in water. Therefore, the linker modification did not significantly affect the secondary structure of the free peptides in the water.

### 3.2. Complexation of Plasmid DNA and Stability of CPP–pDNA Complexes

Next, the complex formation between the CPP and pDNA was investigated by measuring ζ-potential values of NF55 analogues at different CPP to pDNA charge ratios (CR). The change in ζ-potential from the negative to positive indicates that net negatively charged pDNA molecules are successfully encapsulated by the positively charged CPP and the complex has therefore obtained a positive surface charge. The point at which the potential turns positive indicates the amount of the CPP needed to form stable complexes with the pDNA. It has been shown that the surface charge of the formed complexes is also important for the stability of nanoparticles in suspension, and their initial adsorption onto the cell membrane has been associated with higher toxicity [16]. We observed that while the surface charge of the complexes formed between pDNA and the kinked peptides turned positive relatively uniformly at low and narrow-ranged CR values (Figure 3b), the respective ratios for the linear peptides differed from each other noticeably—the smaller the side chain of the linker amino acid of the peptide, the lower CR is required to achieve stable complexes. This could be explained by the steric hindrance of the linker amino acid’s side chain. The bulkier the side chain, the larger the formed nanoparticles are and the larger amount of CPP required to achieve a stable complex formation. This assumption was partially confirmed by measuring the hydrodynamical diameters of the CPP–pDNA complexes. The nanoparticles formed between the NF55 analogues and the pDNA have a hydrodynamical diameter ranging from approximately 130–160 nm, and complexes formed with kinked peptides tend to be slightly smaller than those formed with linear peptides (Appendix A). These results are in good agreement with studies such as [17], showing that peptides, where the sequence is continued via a side chain amino group, are able to form smaller particles with pDNA.

Next, we measured the peptides’ ability to bind and condense pDNA. Therefore, the complex formation process was further examined by using pDNA intercalating dye, which binds to the double-stranded DNA [18]. We observed that all of the peptides packed the pDNA already at the peptide concentration of 10 μM, corresponding to CR1. Interestingly, one of the peptides—NF55-Dap*—formed stable complexes already at a lower concentration than CR1 (Figure 3c). Based on these results, it can be concluded that all of the new CPPs form nanoparticles with pDNA.

For efficient nucleic acid delivery, the CPP has to be able to also protect the cargo. This is especially important in the case of the CPPs designed for in vivo applications, as the CPP should be able to protect the cargo from proteolytic degradation [19]. Therefore, the stability of the CPP–pDNA complexes to the proteolytic degradation was assessed by coincubation with proteinase K, followed by measurement of accessible pDNA. We observed that all the CPPs exerted similar resistance to enzymatic digestion. Interestingly, while the complexes formed with NF55-Dap* are the least resistant to the degradation of proteinase K, NF55-Dap complexes are the most resistant to enzymatic degradation (Figure 3d). Although these two peptides have very similar compositions, they exerted completely different resistance to enzymatic digestion. This indicates that even small modifications on the linker position have a significant impact on the physicochemical properties of the peptides and the particles formed between the CPP and pDNA. To further investigate the stability of NF55 analogues, CPP–pDNA complexes were coincubated with different concentrations of heparin sodium salt. Heparin is a negatively charged polysaccharide, a competitor molecule for negatively charged pDNA. When measuring the amount of liberated pDNA, we observed that the complexes formed with the linear peptides were more stable against heparin displacement than complexes formed with the kinked peptides (Appendix A). This observation is in accord with the proteolytic stability of the NF55-Dap* over the NF55-Dap analogues.

Additionally, we investigated the ability of NF55 analogues to interact with pDNA and limit its migration in agarose gel. Since most of the peptides formed stable complexes with pDNA at CR2, we hypothesized that there would be no significant difference between peptides. However, our observation was that while the linear peptides successfully condensed pDNA into stable nanoparticles, limited its migration in gel, and almost fully protected the pDNA from the NA interacting dye, as indicated by the absence of signal in the gel tooth, there was some pDNA migration was visible for the kinked peptides (Appendix A). Additionally, the ability of the kinked peptides to limit pDNA migration increased with the length of the linker. Although the stability of nanoparticles formed with the linear peptides has its advantage while delivering pDNA, the release of the pDNA cargo is ultimately required for the gene expression. Based on the gel migration assay, we may hypothesize that the kinked peptides have an advantage in the latter.

The nanoscopic structure of the CPP–pDNA complexes was also analyzed using atomic force microscopy combined with infrared nanospectroscopy (AFM-IR). This AFM-based infrared technique allows simultaneous determination of the topographical characteristics of the sample and obtaining vibrational spectroscopy information of individual particles with nanoscale spatial resolution. The topographical images obtained from complexes prepared with the NF55 analogue peptides reveal the extensive presence of globular aggregates with dimensions within the nanoscopic range, consistent with sizes and polydispersity indexes found in DLS assays (Appendix A). However, it is also possible to observe the presence of rounded particles with lateral dimensions of around 50 nm, which is in close agreement with the dimensions of complexes produced with other CPPs of the NickFect family [6,7]. In addition to these very small structures, it is also possible to observe the coexistence of domains with dimensions that reach about 200 nm. A detailed inspection of the cross section of these domains reveals the presence of roughness that extends for several tens of nanometers, indicating that the domains are composed of smaller subunits and are probably the result of the aggregation of complexes (Appendix A). The vibrational profiles of these aggregates also reveal some structural diversity, and infrared data confirm that although the peptides are highly similar in terms of composition, the replaced linker amino acid has a significant impact on the chemical bonding behaviour of the complexes (Appendix A). For example, the amide I bands of the complexes formed with CPPs containing Dab and Lys as linker amino acids display greater complexity than those prepared with CPPs containing Dap and Orn residues, indicating that these peptides interact differently with pDNA in the formation of complexes. In most of the analyzed regions, it is possible to notice a strong peak around 1655 cm^−1^ which, in combination with information revealed by CD experiments, can be assigned to alpha-helix or random-coil conformations [20,21]. The appearance of resonances in the 1665–1670 cm^−1^ range suggests that β-turn secondary structures appear upon complexation between CPPs and pDNA. The β-turns generally correspond to changes in direction and loops along the peptide chains, and here it can be correlated with the high degree of packaging and condensation resulting from the formation of complexes between the CPPs and the DNA chains [22]. Finally, it has been observed that complexation with nucleic acids may induce some degree of β-sheet content, as evidenced by weak resonances in the range of 1630–1640 cm^−1^ and near 1690 cm^−1^. This effect is especially notable for NF55, which exhibits higher hydrophobicity compared to the other CPPs.

### 3.3. Transfection of CHO-K1 and A549 Cells

We next assessed the transfection efficiency of the NF55 linker analogues in the CHO-K1 cells. We observed that in the case of the linear peptides, the transfection efficiency in FM (media containing 10% FBS) decreases with an increase of methyl groups in the linker amino acid side chain (Figure 4a). This indicates that the bulkiness of the side chain may hinder transfection efficiency. Contrary to the linear peptides, the increase of methyl groups in the linker seems to have almost no effect on the transfection efficiency of the kinked peptides (Figure 4a). Surprisingly, any modifications in the linker position have no noticeable effect on transfection efficiency in SFM (serum free media) (Figure 4b). We also observed that the transfection efficiency of NF55-Dap* at CR1 is noticeably higher than for other peptides at the same CR value. This may be due to the of NF55-Dap* to form stable nanoparticles even at a very low peptide concentration, as indicated in Figure 3b,c.

In addition to the transfection efficacy measured by total gene expression in the cell population, another important transfection efficacy parameter is the number of transfection-positive cells. We estimated this by transfecting pEGFP and calculating the percentage of cells expressing green fluorescent protein by using flow cytometry. For the linear peptides, our results are in good accordance with the luciferase assay results and show that transfection efficiency decreases with an increase of linker side chain bulkiness in FM (Figure 4c). Interestingly, and different from what we observed with the luciferase assay, the same trend of reduced transfection with a larger side chain was observed in SFM. Although we did not observe differences in transfection with pDNA between the kinked peptides, the length of the linker seems to be important while transfecting the cell population. As indicated by our results displayed in Figure 4c, the shorter the linker, the higher the transfection efficiency in FM. However, we hypothesize that more space between the N- and C-terminus of the CPP may have an advantage in SFM.

As high transfection efficiency is often associated with increased cellular toxicity [7], we also evaluated the safety of the complexes formed between NF55 analogues and pDNA by using the MTS proliferation assay, which reflects the metabolic state of the cells after transfection. In order to exacerbate any differences in inducing unwanted side effects, the experiment was conducted in the harshest possible conditions: a serum-free environment without the change of media during incubation of the cells with the CPP–pDNA complexes. This allows the obtainment of more pronounced results since the addition of serum generally drastically reduces the possible effects on the cells [23]. We observed that the designed peptides affected the cell viability significantly less than the commercial transfection reagent PEI MAX (N/P ratio 20 used). However, it appeared that the CPPs that were shown to achieve a high transfection efficacy (the kinked NF55 analogues) affected cell proliferation more than the linear NF55 analogues (Figure 4d).

It has been shown that complexes formed between NF55 and pDNA lead to high reporter signal levels in the lungs [8]. Therefore we investigated whether the peptides can deliver pDNA into the lung cancer cell line A549. This is a human lung adenocarcinoma cell line that has been widely used in cancer research [24,25,26]. The first difference we observed between the total reporter levels in CHO-K1 and A549 cells was that in the case of A549, CR2 resulted in higher reporter values compared to CR3 for most of the peptides in both FM and SFM (Appendix A). The kinked peptides had a higher transfection efficiency than their linear analogues in SFM. However, in FM, the NF55-Dap had the highest transfection efficiency, followed by the NF55-Dab. We also observed that in FM, transfection efficiency was higher the shorter the linker and the shorter the side chain of the linker peptide. However, in SFM, this trend did not apply. What is more, all of the tested peptides except for NF55-Orn and NF55-Lys had higher transfection efficiency compared to the commercially used transfection reagent PEI MAX (Figure 4e,f). Based on the results obtained from cell culture transfection experiments, it can be concluded that several NF55 analogues are potentially suitable for in vivo transfection.

### 3.4. Assessment of Transfection Efficiency of NF55 Analogues In Vivo

Having obtained proof that NF55 analogues have a great potential for delivering pDNA into the cells, the next step was to assess the efficiency of the designed NF55 analogues for pDNA delivery in vivo in a mouse model. We thus quantified luciferase after a single i.v. injection by live animal imaging and quantitation from postmortem tissue homogenates. Based on the IVIS images, the highest reporter signal was detected in regions corresponding to the spleen, the liver, and the lungs. A high reporter signal was detected in almost all of the groups especially the complexes formed between pDNA and NF55-Dap or NF55-Dab* having high luciferase expression in the lungs and liver (Figure 5a). The complexes with NF55-Orn and NF55-Lys did not induce detectable luciferase expression and stayed at the same level with naked pDNA which was used as a control. The low luciferase expression of NF55-Orn and NF55-Lys is in accord with the results obtained when transfecting the cell culture, where both CPPs were significantly less efficient than other tested peptides in both CHO-K1 and A549 cells.

These results were further confirmed by quantifying luciferase levels from the homogenized tissues (Figure 5b). The spleen, liver, and lung were chosen as the tissues to be analysed based on the IVIS results and as an expression of reporter signal has been detected from these tissues historically when using NF55 as a transfection reagent [8]. Based on tissue homogenate analysis, the highest expression was induced in the lung tissues, as reported previously for NF55 (Appendix A) [8]. The kinked peptides were especially able to efficiently deliver pDNA into the tissues with NF55-Dab* being the most efficient out of all the peptides in all of the tissues examined. Aside from NF55-Dap and NF55-Dab, the linear peptides were not as efficient for the delivery of pDNA as reporter levels achieved in all of the tissues examined were comparable to the naked pDNA. These results are once again in good accordance with what we observed when transfecting the cell culture in vitro: longer side chains of the linear peptides are accompanied by a decrease in transfection efficiency (Figure 4a,c,e).

Additionally, the safety of NF55 analogues which led to the highest expression level in vivo was evaluated by measuring alanine aminotransferase (ALAT) and aspartate aminotransferase (ASAT) levels from the blood of mice injected with complexes formed between NF55 analogues and pDNA nanoparticles. The body weights of the mice injected with CPP–pDNA complexes were also measured preinjection and prior to the sacrifice. Based on the obtained results, no significant toxicity was observed for the NF55 analogues included in the experiment (Appendix A).

## 4. Conclusions

In the current work, we aimed to improve the methods available for the efficient delivery of plasmid DNA cargo in cell cultures, especially in vivo. We developed several unique CPPs that exert potential biological effects. Our two new CPPs, NF55-Dap and NF55-Dab*, exerted a significant boost on gene delivery, reflected in the enhanced transgene expression in several tissues. Based on this, these new CPPs have a high potential for gene delivery applications, especially for the treatment of lung diseases.

## Figures and Tables

**Figure 1 pharmaceutics-15-00883-f001:**
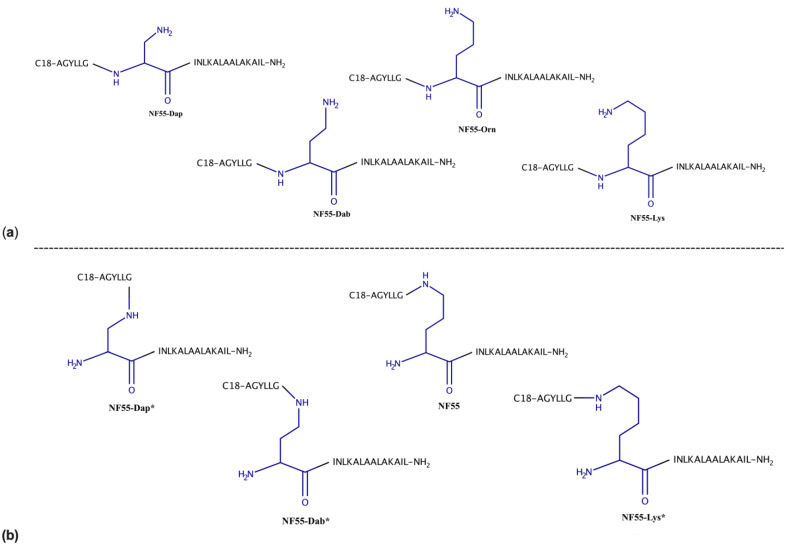
NF55 analogues with different linker amino acids, where (**a**) synthesis is continued through ⍺-NH_2_ group of the linker amino acid (the linear peptides); (**b**) synthesis is continued through the side chain amino group of the linker amino acid (the kinked peptides).

**Figure 2 pharmaceutics-15-00883-f002:**
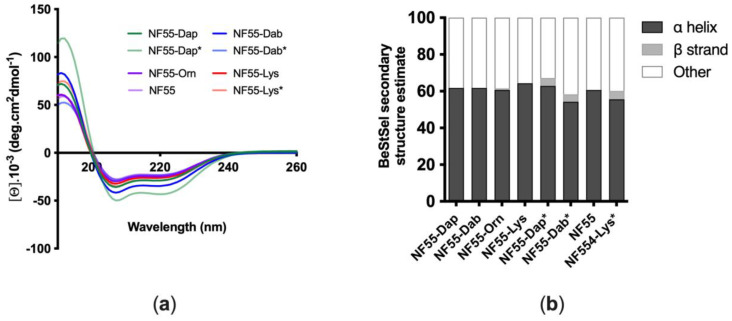
Conformational analysis of NF55 analogues by circular dichroism spectroscopy. (**a**) CD spectra of NF55 analogues; (**b**) secondary structures of NF55 analogues in water.

**Figure 3 pharmaceutics-15-00883-f003:**
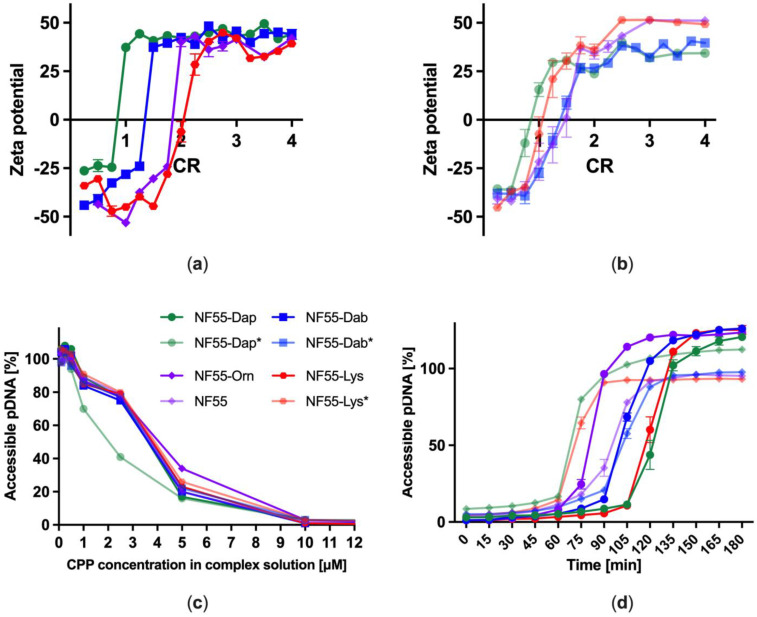
(**a**) ζ-potential values of linear and (**b**) kinked NF55 analogues at different CR values; (**c**) Quantification of accessible pDNA at different CPP concentrations; (**d**) CPP–pDNA complex resistance to enzymatic digestion by Proteinase K.

**Figure 4 pharmaceutics-15-00883-f004:**
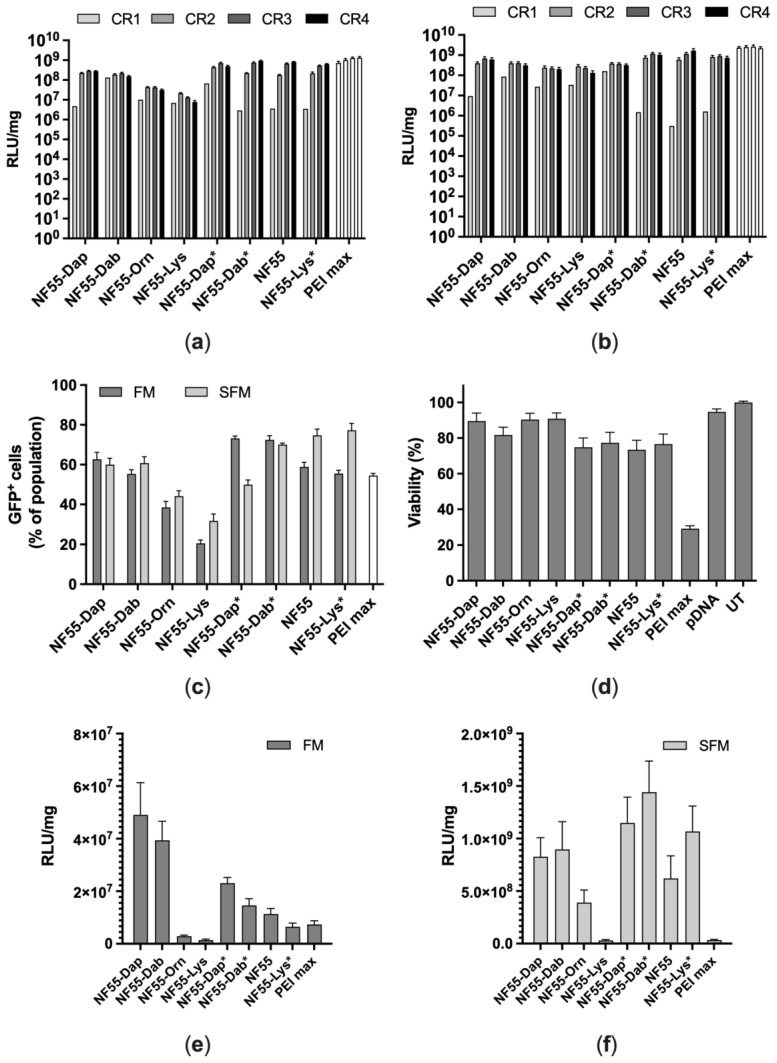
(**a**) Transfection of CHO-K1 cells with NF55 analogues at different CR values in FM and (**b**) SFM. (**c**) The ability of NF55 analogues to transfect cell population was investigated by counting transfected cells using flow cytometry. (**d**) Evaluation of cell viability by MTS cell proliferation assay. (**e**) Transfection of A549 cells with NF55 analogues at CR2 in FM and (**f**) SFM.

**Figure 5 pharmaceutics-15-00883-f005:**
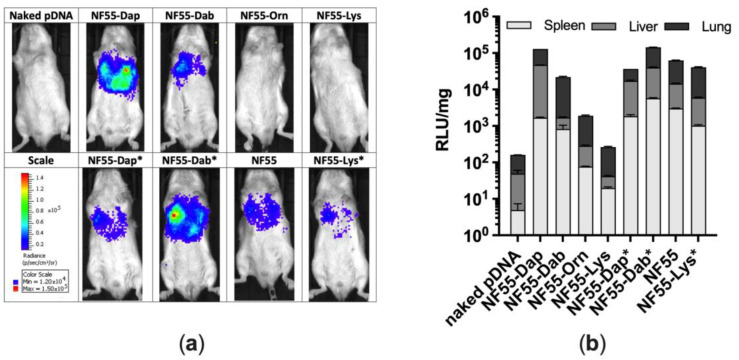
(**a**) Bioluminescence live animal imaging using IVIS; (**b**) Luciferase levels measured from homogenized tissues.

## Data Availability

All data is available upon reasonable request from the corresponding author.

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
