# Peer review of "Modification of the Linker Amino Acid in the Cell-Penetrating Peptide NickFect55 Leads to Enhanced pDNA Transfection for In Vivo Applications"

_pharmaceutics, 2023, doi:10.3390/pharmaceutics15030883_

Round 1
Reviewer 1 Report
The manuscript entitled “Modification of the Linker Amino Acid in the Cell-Penetrating Peptide
NickFect55 leads to Enhanced pDNA Transfection for in vivo Applications” by Heleri H. Härk et al. investigated cell penetration peptides as a potential drug delivery platform for in vivo applications. A series of peptides were first designed and prepared with different modified linkers and charge distribution. Then these peptides were complexed with plasmid DNA for characterization studies including CD spectroscopy, zeta-potential, AFM-IR and encapsulation measurement. Finally, in vitro and in vivo functional studies were conducted to study the delivery efficacy as well as safety profile of these CPP-pDNA. Overall, the experiments in the manuscript are well designed with proper controls and provide a relative comprehensive evaluation of properties and functions of the peptide-based drug delivery platform. However, following questions still need to be addressed before publication:
1. In Introduction section, the authors should introduce more of current research progress for nucleic acids delivery. For example, several GalNAc conjugated siRNAs have been approved by FDA for hepatic delivery while extrahepatic delivery remains challenging in the field. Regarding CPP, what’s the major advantage over other delivery systems? Any CPP has been well studied or even in clinical trial stage? Please expand this section to provide more background info.
2. In Figure 3d, the authors evaluated the CPP/pDNA stability in Proteinase K condition. However, nuclease degradation should be the major problem for unprotected pDNA. Have the authors studied the nuclease stability of CPP/pDNA?
3. It’s very interesting that the transfection efficiency of linear peptides reduced with increasing methyl groups in 10%FBS media condition, but no such effect shown in kinked peptides. Is it possibly attributed to different protein binding profile in 10%FBS media? Increasing methyl groups may increase hydrophobicity and leads to higher protein binding, which may be also affected by peptide structure such as linear or kinked. Running a quick gel assay of different peptides in 10%FBS media may be worthy trying.
4. The authors use “NF55-Orn*” in Figure 4d-f, different from “NF55” in Figure 4a-c. Are they the same peptide? If so, please unify the name.
5. Figure 4e-f compared the transfection efficiency of pDNA using different CPPs at CR2 in A549 cells. However, the ranking order of transfection efficiency looks very different from that in CHO-K1 cells shown in Figure 4a-b. Why the behaviors of these peptides are so different in different cell types? Any known receptors mediating CPP cellular uptake?
6. For in vivo bioluminescent study, how many mice tested per group? Have the authors considered studying it at different time point? I suggest removing Figure 5c-d or moving into supplementary material because the % calculation here is not real and accurate since only three types of organs taken into analysis.
Reviewer 2 Report
In this manuscript, the authors developed several unique CPPs NF55 analogues, which showed excellent gene transfection efficacy in vitro. In addition, the NF55-Dap/pDNA complex and NF55-Dab*/pDNA complex showed highest transfection efficacy in vivo. Meanwhile, these NF55 analogues exhibited low toxicity in vivo, which was important for gene delivery application. The manuscript is presented in a logical and systematic manner. The research is interesting, and the manuscript was well written. In addition, the experimental section is well described, and the results are well discussed. It’s positively recommended published in Pharmaceutics after minor revision:
1. The positively charged CPPs was able to interaction with negatively charged cell membranes, which exhibited high cell cytotoxicity. In this work, the NF55/pDNA complexes also showed high zeta potential values, which maybe lead to potential threat. How to avoid the toxicity of the complexes in vitro and in vivo?
2. The authors evaluated the in vitro gene transfection efficacy of NF55/pDNA complexes in FM (media containing 10% FBS) and SFM (serum free media). Could the authors provide us the reasons for choosing different media?
3. In Figure 5a, the luciferase expressions of NF55-Orn and NF55-Lys groups were very low, and the fluorescence was undetectable. Why? The authors could provide us more discussion in Page 12, Lines 444-445.
